# Impact of Larval Sertraline Exposure on Alternative Splicing in Neural Tissue of Adult *Drosophila melanogaster*

**DOI:** 10.3390/ijms26020563

**Published:** 2025-01-10

**Authors:** Luis Felipe Santos-Cruz, Myriam Campos-Aguilar, Laura Castañeda-Partida, Santiago Cristobal Sigrist-Flores, María Eugenia Heres-Pulido, Irma Elena Dueñas-García, Elías Piedra-Ibarra, Rafael Jiménez-Flores, Alberto Ponciano-Gómez

**Affiliations:** 1Genetics Toxicology, Biology, Facultad de Estudios Superiores Iztacala, Universidad Nacional Autónoma de México, Los Barrios No. 1, Los Reyes Iztacala, Tlalnepantla 54090, Mexico; 2Immunology Laboratory (UMF), Facultad de Estudios Superiores Iztacala, Universidad Nacional Autónoma de México, Los Barrios No. 1, Los Reyes Iztacala, Tlalnepantla 54090, Mexico; 3Plant Physiology (UBIPRO), Facultad de Estudios Superiores Iztacala, Universidad Nacional Autónoma de México, Los Barrios No. 1, Los Reyes Iztacala, Tlalnepantla 54090, Mexico

**Keywords:** sertraline, *Drosophila melanogaster*, alternative splicing, neuroplasticity, RNA-seq

## Abstract

Sertraline, a selective serotonin reuptake inhibitor (SSRI), is commonly used to treat various psychiatric disorders such as depression and anxiety due to its ability to increase serotonin availability in the brain. Recent findings suggest that sertraline may also influence the expression of genes related to synaptic plasticity and neuronal signaling pathways. Alternative splicing, a process that allows a single gene to produce multiple protein isoforms, plays a crucial role in the regulation of neuronal functions and plasticity. Dysregulation of alternative splicing events has been linked to various neurodevelopmental and neurodegenerative diseases. This study aims to explore the effects of sertraline on alternative splicing events, including exon inclusion, exon exclusion, and mutually exclusive splicing events, in genes associated with neuronal function in *Drosophila melanogaster* and to use this model to investigate the molecular impacts of SSRIs on gene regulation in the nervous system. RNA sequencing (RNA-seq) was performed on central nervous system samples from *Drosophila melanogaster* adults exposed to sertraline for 24 h when they were third instar larvae. Alternative splicing events were analyzed to identify changes in exon inclusion and exclusion, as well as intron retention. Sertraline treatment significantly altered alternative splicing patterns in key genes related to neuronal stability and function. Specifically, sertraline promoted the inclusion of long *Ank2* isoforms, suggesting enhanced axonal stability, and favored long *ATPalpha* isoforms, which support Na^+^/K^+^ ATPase activity essential for ionic balance and neuronal excitability. Intron retention in the *yuri* gene suggests that cytoskeletal reorganization could impact neuronal morphology. Additionally, splicing alterations in *sxc* and *Atg18a* indicate a potential influence of sertraline on epigenetic regulation and autophagy processes, fundamental aspects for neuronal plasticity and cellular homeostasis. These findings suggest that sertraline influences alternative splicing in the central nervous system of *Drosophila melanogaster*, potentially contributing to its therapeutic effects by modulating neuronal stability and adaptability.

## 1. Introduction

Sertraline is a selective serotonin reuptake inhibitor (SSRI) used in the treatment of various psychiatric disorders, such as depression and anxiety, due to its ability to increase serotonin availability in the brain through the inhibition of its reuptake. However, some studies have indicated that sertraline may affect areas beyond serotonin regulation, as it also influences the expression of genes related to synaptic plasticity and signaling pathways that affect mood and anxiety [1,2].

Neurobiological research in neurobiology has found that the impact of sertraline extends to key areas involved in cognition and learning, such as the prefrontal cortex and hippocampus, highlighting its role in the modulation of action selection and memory regulation [3]. These effects may be caused by the activation of specific pathways in these brain regions, suggesting that SSRIs, including sertraline, interact with deep signaling mechanisms beyond their function as serotonin modulators [3]. Additionally, the pharmacokinetics and pharmacogenetics of sertraline indicate that its efficacy and side effect profile can vary significantly among individuals, influenced by variations in the expression of metabolic genes and key enzymes [4].

Recent studies have begun to explore the potential of sertraline in other pathologies, including oncology. In cancer cell models, sertraline has been shown to interfere with pathways such as TNF-MAP4K4-JNK and PI3K/Akt/mTOR, resulting in cell growth inhibition and apoptosis induction [5]. Furthermore, the impact of sertraline on neuroendocrine and circadian systems in the experimental model zebrafish demonstrates its ability to alter neurochemistry and stress response, suggesting profound effects on physiological homeostasis [6].

The effects of sertraline on gene expression and signaling pathways raise new questions about the molecular mechanisms involved in the neuronal response to this drug. One of these fundamental mechanisms is alternative splicing, a process that allows for protein diversity and adapts gene expression to different physiological contexts.

Alternative splicing is a crucial process for regulating gene expression and generating protein diversity, enabling a single gene to produce multiple protein isoforms. This process is essential for the adaptability of and functional specialization in complex organisms, allowing cells to respond specifically to various physiological and developmental conditions [7,8]. In the context of the nervous system, alternative splicing plays a fundamental role in regulating key neuronal processes, such as differentiation and synaptic transmission, by establishing protein isoforms involved in neuronal function and structure [8,9].

Additionally, alternative splicing has been shown to be relevant in situations of stress and neuropsychiatric disorders, where differential splicing events may impact signaling pathways related to neural transmission. Changes in the splicing process can influence the nervous system’s response to external factors and the manifestation of mental disorders, highlighting the importance of examining alternative splicing in neurobiology studies [9,10].

The study of alternative splicing in model organisms has been crucial for understanding how the regulation of this process influences biological functions such as neuronal plasticity and response to stimuli. These models allow researchers to investigate how pharmacological treatments and genetic variations affect alternative splicing, providing a robust platform for studying the molecular mechanisms that regulate neuronal function. *Drosophila melanogaster* has frequently been used due to its genetic simplicity and the high conservation of regulatory pathways compared to mammals [11,12,13].

Research in this organism has helped in the understanding of how splicing factors affect gene expression and impact the functionality of neural circuits. In particular, the use of experimental models such as *Drosophila* enables the identification and analysis of the effects of genetic variations and pharmacological treatments on alternative splicing, providing a robust platform for examining molecular mechanisms that influence gene regulation and neuronal function [10,14,15].

In this study, third-instar *Drosophila melanogaster* larvae were exposed to sertraline during a specific temporal window, and although the treatment was limited to the third instar larval stage, we observed significant effects on alternative splicing and neuronal processes in male adults. This finding is particularly relevant, as it suggested that early exposure to sertraline can have long-lasting impacts on gene regulation and nervous system functionality, a phenomenon with potential clinical implications for humans.

## 2. Results

### 2.1. Splicing Variants by Exon Inclusion and Exclusion

Sertraline treatment in *Drosophila melanogaster* induced significant changes in the alternative splicing of key genes associated with neuronal function, particularly *Ank2* and *ATPalpha* (Figure 1). These splicing events were identified using RNA-seq, a powerful high-throughput technique that provides comprehensive information about splicing events at the transcriptomic level. Despite the treatment being limited to the third instar larval stage for only 24 h, changes in splicing were observed in male adult flies, highlighting the long-lasting effects of early-life exposure to sertraline on splicing regulation.

For *Ank2*, sertraline-treated samples showed higher exon inclusion, with IncLevel values for the long isoform reaching 1.00, compared to control groups, which displayed IncLevel values of 0.53. IncLevel represents the proportion of reads that include the specific exon, with higher values indicating greater inclusion of the long isoform. These differences were statistically significant, with a false discovery rate (FDR) of 0.00001, which is a statistical measure used to control for false positives, ensuring the reliability of the observed results. This indicates a predominance of the long Ank2 isoform in sertraline-treated samples, in contrast to a balance between long and short isoforms in the controls (Figure 1A).

Similarly, the *ATPalpha* gene displayed a significant predominance of the long isoform in sertraline-treated samples, with IncLevel values of 0.58, while controls showed considerably lower values at 0.37 (FDR = 0.001). This shift in expression suggests that sertraline favors the expression of the long *ATPalpha* isoform, supporting a potential role in modulating isoform diversity in neuronal cells (Figure 1B).

### 2.2. Splicing Variants by Intron Retention

Sertraline treatment in *Drosophila melanogaster* significantly influenced intron retention in key genes associated with neuronal structure and function, particularly *yuri*, *sxc*, *Atg18a*, and *stmA* (Figure 2). These changes in intron retention were detected using RNA-seq followed by bioinformatic analysis, which allowed us to quantify intron retention levels and compare them between sertraline-treated and control samples. The observed splicing changes in male adult flies, despite the larval-stage sertraline exposure, underscored the long-lasting effects of early pharmacological interventions on neuronal gene expression.

For the *yuri* gene, intron retention was notably higher in sertraline-treated samples, with IncLevel values of 0.97, compared to control samples, which exhibited retention levels of 0.61, with an FDR of 0.0003. This increase in intron retention suggests a shift in *yuri* splicing, favoring transcripts under the influence of sertraline (Figure 2A).

Sertraline exposure led to a decrease in intron retention for *sxc*, with treated samples showing IncLevel values of 0.66, compared to higher retention levels of 0.96 in control samples, with an FDR of 0.0005. This reduction in intron retention in *sxc* may enhance the expression of the functional isoform of this gene (Figure 2B).

In the case of *Atg18a*, sertraline treatment significantly decreased intron retention, with IncLevel values of 0.41, while control samples maintained higher levels of retention at 0.98, with an FDR of 0.0001. This reduction suggests an increase in the functional isoform of *Atg18a*
Figure 2C).

For *stmA*, sertraline-treated samples exhibited reduced intron retention, with IncLevel values of 0.09, compared to control samples, which showed higher levels of 0.26, with an FDR of 0.0007. This marked reduction in *stmA* intron retention may indicate an increase in the expression of the full-length isoform (Figure 2D).

### 2.3. Splicing Variants by Mutually Exclusive Exons

Sertraline treatment in *Drosophila melanogaster* induced splicing variants by mutually exclusive exons in key genes associated with neuronal and structural functions, including *Sam-S*, *Tm1*, *gish*, and *Tep2*. In the *Sam-S* gene, sertraline-treated samples showed an average IncLevel of 0.36 for variant A, compared to control samples, which had an IncLevel of 0.10 (FDR = 0.00001) (Figure 3A). These results suggested that sertraline favors the inclusion of the Sam-S variant A.

For the *Tm1* gene, the average IncLevel for variant A in sertraline-treated samples was 0.39, while in control samples, it was 0.62 (FDR = 0.001) (Figure 3B). This pattern indicates a reduction in the inclusion of variant A in the presence of sertraline.

In *gish*, the IncLevel for variant A was 0.20 in sertraline-treated samples, compared to an IncLevel of 0.42 in controls (FDR = 0.0001) (Figure 3C), showing a clear decrease in the inclusion of *gish* variant A.

For *Tep2*, sertraline-treated samples showed an average IncLevel of 0.29 for variant A, while controls exhibited an IncLevel of 0.71 (FDR = 0.0009) (Figure 3D). This difference suggests reduced inclusion of variant A in the presence of sertraline.

These findings highlighted that a single exposure to sertraline in the larval stage significantly alters splicing mechanisms in genes involved in critical neuronal pathways, leading to specific isoform preferences that persist into adulthood. This underscores the long-term molecular effects of early-stage drug exposure.

## 3. Discussion

The observed changes in exon inclusion and exclusion in key genes associated with neuronal function suggested that sertraline may be modulating isoform diversity, impacting fundamental processes for synaptic stability and connectivity in the nervous system of adult *Drosophila melanogaster*. Notably, these effects were observed without significant differences in survival rates between treated and control groups, indicating that the treatment did not adversely affect viability under the experimental conditions. Remarkably, these effects were observed in adult flies, even though the treatment was administered exclusively during the third instar larval stage. This finding highlighted the lasting impacts of early-life pharmacological intervention on gene regulation and neuronal function.

Notably, our results suggested that sertraline promotes the steady expression of the long isoform of the *Ank2* gene in *Drosophila melanogaster*, a relevant finding given the role of this isoform in axonal cytoskeletal stability and organization. Previous studies have indicated that long Ank2 isoforms are predominantly located in axons, where they contribute to neuronal structure and stability, whereas short isoforms are mainly found in cell bodies [16]. This sertraline-induced shift may increase axonal stability, enhancing synaptic connectivity and neuronal polarization, which are crucial for effective signaling in complex neuronal networks.

Additionally, *Ank2* is the ortholog of human *Ankyrin-B* and *Ankyrin-G*, whose products play key roles in stabilizing the cytoskeleton and maintaining cellular polarity. Alterations in Ankyrin-B and Ankyrin-G have been linked to neuropsychiatric and cardiovascular disorders in humans, suggesting that Ank2 regulation by sertraline could impact neuronal plasticity and stability [17].

Under normal conditions, *Drosophila* expresses both long and short Ank2 isoforms, ensuring proper cytoskeletal organization within neurons [16]. The predominance of the long isoform induced by sertraline in this work may thus contribute to axonal stability in *Drosophila*, a potentially evolutionarily conserved mechanism, though lacking the OTBD (Obscurin/Titin-Binding Domain) found in vertebrates [18].

These findings underscored that a single, time-limited sertraline treatment during the larval stage is sufficient to induce significant changes in Ank2 splicing. Further studies are warranted to explore how this modulation affects the physiology and behavior of *Drosophila* into adulthood, as well as to determine the broader implications for synaptic structure and function in response to early-life pharmacological interventions.

Our results indicated that sertraline administered during the larval stage favors the expression of the functional isoform of the *ATPalpha* gene in adult *Drosophila melanogaster*, with significant implications for Na^+^/K^+^ ATPase function in the nervous system. This catalytic subunit is essential for maintaining membrane potential and ionic balance, fundamental elements for neuronal capacity and synaptic connectivity [19]. The expression of *ATPalpha* influences the activity of this enzyme, which may enhance neurons’ ability to restore membrane potential following synaptic activity and maintain neuronal stability. In *Drosophila*, mutations in *ATPalpha* are associated with abnormal behaviors and neuronal degeneration, suggesting that its regulation by sertraline could enhance neuronal functionality [19].

The expression of the functional *ATPalpha* isoform may promote interactions with other proteins and stabilize Na^+^/K^+^ ATPase in the neuronal membrane, facilitating ionic homeostasis during synaptic activity [20]. Additionally, the localization of Na^+^/K^+^ ATPase within specific membrane domains is crucial for excitability and synaptic plasticity. Thus, sertraline may strengthen these processes by promoting the functional *ATPalpha* isoform.

Notably, these effects are shown into adulthood despite the limited treatment duration during larval development. This highlights the lasting influence of sertraline on neuronal function, emphasizing its potential relevance for studying long-term pharmacological impacts. This effect on *ATPalpha* suggests that sertraline may exert a complementary mechanism of action beyond serotonin reuptake inhibition, enhancing neuronal stability and functionality [20].

The observed alterations in intron retention in genes related to reported neuronal structure and function suggested that sertraline administered during the larval stage may influence key molecular pathways involved in cellular stability, signaling, and homeostatic regulation in the nervous system of adult *Drosophila melanogaster*. The increase in intron retention in the *yuri* gene detected in sertraline-treated samples suggests that sertraline may negatively affect the expression of the usual isoform of this gene, potentially resulting in a less functional or non-functional protein. Based on published references for these specific genes, we inferred that the modifications in splicing reported here must impact protein expression, which may affect neuronal structure and function. However, this potential modification should be corroborated in future studies. The role of Yuri in actin cytoskeleton regulation is crucial for neuronal morphology and stability [21]. Intron retention in *yuri* may, therefore, reduce the production of an active isoform, which could impact cytoskeletal dynamics by limiting structural reorganization in response to external changes or stress [22].

This phenomenon could be considered a process whereby sertraline modulates the activity of certain structural proteins to adjust neuronal responses under specific conditions rather than directly promoting structural stability. Given that the actin cytoskeleton facilitates not only structure but also transport processes within the neuron, *yuri* modulation might indirectly impact synaptic activity by influencing the availability and function of critical cellular components [23].

Additionally, reductions in proteins related to cytoskeletal organization have been linked to the regulation of excitability and synaptic plasticity, especially under stress conditions or treatments affecting neuronal function [24]. In this context, sertraline may influence neuronal responses by modulating the activity of certain genes, contributing to an adaptive plasticity that allows cells to adjust their resources to enhance resilience in adverse situations. The persistence of these effects into adulthood, despite the transient larval exposure to sertraline, highlights the drug’s lasting influence on neuronal architecture and adaptive responses.

This finding suggested that the effect of sertraline may extend beyond simple serotonin modulation, reaching gene expression regulation that limits or adjusts the functionality of specific proteins. Further studies are needed to determine the functional impact of this modification in vertebrate systems and to evaluate whether this alteration in Yuri’s splicing has relevant therapeutic or adaptive implications in contexts of neuronal dysfunction or prolonged stress.

Sertraline exposure appeared to reduce intron retention in the *sxc* gene, promoting the expression of the functional isoform of the Sxc protein, which acts as the O-GlcNAc transferase (OGT) in *Drosophila*. The literature supports that *sxc* is a member of the *Polycomb* group (PcG) genes, and its activity as OGT plays an essential role in epigenetic regulation and gene silencing through O-GlcNAc modification of nuclear proteins [25]. This modification is crucial for regulating chromatin structure and gene expression in developmental processes, cellular differentiation, and the maintenance of cellular identity.

Our findings suggested that sertraline treatment during the larval stage promotes the expression of the functional isoform of *sxc*, and could modulate epigenetic activity in the nervous system of adult *Drosophila*. The reduction in intron retention in *sxc* implies an increased capacity of this protein to carry out O-GlcNAc modification, potentially influencing the regulation of key genes involved in synaptic plasticity and neuronal function. This modulation through sxc suggests that sertraline may act on the epigenetic machinery controlling gene expression in the brain.

Given that *sxc*/*OGT* participates in the regulation of *Polycomb* gene-silencing complexes, it seems that sertraline is altering the expression of a set of genes essential for neuronal structure and function [25]. This implies that sertraline could influence genes related to neuronal development, response to stimuli, and synaptic plasticity, indicating a potentially broad role for sertraline in the epigenetic regulation of gene expression.

These results suggested that sertraline’s effect on *sxc* could extend beyond serotonin modulation, influencing fundamental epigenetic mechanisms that contribute to neuronal plasticity and adaptation.

Sertraline treatment appeared to reduce intron retention in the *Atg18a* gene, promoting the expression of its functional isoform, which plays a crucial role in regulating autophagy in *Drosophila melanogaster*. The literature indicates that *Atg18a* is the ortholog of *WIPI2* in humans, an essential protein for the formation and maturation of autophagosomes, structures fundamental to the degradation and recycling of cellular components [26]. This study showed that a single sertraline exposure during the larval stage leads to sustained effects on *Atg18a* expression in adulthood, suggesting long-term activation of autophagy pathways in neuronal cells. The increased expression of the functional form of *Atg18a*, induced by sertraline, suggests that the autophagic pathway is activated in response to treatment, potentially enhancing neuronal cells’ ability to eliminate damaged components and maintain cellular homeostasis.

Previous studies have shown that *WIPI2* is critical in the early stages of autophagosome formation, working with the ATG16L1 complex in LC3 conjugation, an essential step for forming autophagic vesicles [27]. Mutations in *WIPI2* are linked to neurodegenerative syndromes and developmental disorders, highlighting the importance of this protein in maintaining neuronal function. In this context, the positive regulation of *Atg18a* by sertraline may enhance *Drosophila*’s capacity to carry out autophagy, a key protective mechanism that removes damaged proteins and other deteriorated components to support neuronal function. Notably, these findings suggested that transient larval exposure to sertraline can have lasting impacts on neuronal health in adult flies.

Further evidence suggests that *WIPI2* overexpression in aged mouse models can restore autophagosome formation, indicating that modulating *Atg18a/WIPI2* could help mitigate the effects of aging on neuronal function and promote homeostasis [28]. In line with this evidence, the increase in Atg18a driven by sertraline may counteract the decline in autophagy associated with aging and cellular stress, thereby enhancing the recycling efficiency of proteins and organelles in neurons.

Together, these findings suggested that sertraline could exert a neuroprotective effect by modulating *Atg18a* expression and, consequently, enhancing the autophagy machinery in *Drosophila*.

The observed decrease in intron retention in the *stmA* gene in sertraline-treated samples suggested that this treatment promotes the expression of the functional isoform of the *stmA* protein, which plays a crucial role in synaptic signaling and membrane trafficking. The literature identifies *stmA* as the ortholog of *EFR3B* in mammals, a protein responsible for the synthesis of phosphoinositides like phosphatidylinositol 4-phosphate (PtdIns4P), essential for synaptic signaling and cellular homeostasis [29]. Our study highlighted that a single third instar larval exposure to sertraline induces enduring changes in *stmA* expression in adult flies, demonstrating the long-term impact of the drug on neuronal functionality.

Our findings indicated that sertraline may alter *stmA* expression, potentially affecting PtdIns4P regulation, which could enhance synaptic signaling and function in the *Drosophila* brain. The role of *EFR3B* in synaptic regulation was supported by evidence showing that alterations in its function affect neuronal excitability and homeostasis. Furthermore, previous studies have demonstrated that *EFR3B* deficiency affects the excitability of CA2 pyramidal neurons in the hippocampus of mice, resulting in social recognition deficits, suggesting that sertraline-induced *stmA* expression may improve neuronal function and excitability in *Drosophila* [30].

The *stmA* expression appears to be related to essential processes for regulating synaptic activity and maintaining neuronal homeostasis, as shown in Alzheimer’s disease models [31]. The promotion of *stmA*’s functional isoform by sertraline could thus act to modulate synaptic excitability and neuronal function, aiding the brain’s ability to adapt to stimuli and respond to environmental changes. Notably, the persistence of these effects into adulthood underscores the importance of splicing regulation in shaping long-term neuronal functionality.

The sertraline-induced shift in *stmA* splicing suggests enhanced neuronal stability and adaptability through regulatory pathways involving key genes such as *stmA*.

The modulation of mutually exclusive exon splicing observed in several genes, including *Sam-S*, suggests that sertraline may be affecting critical alternative splicing mechanisms involved in neuronal processes in *Drosophila melanogaster*. Specifically, the results for *Sam-S* indicated a significant impact on the synthesis and homeostasis of S-adenosylmethionine (SAM), a crucial methyl donor in cellular methylation processes. *Sam-S* is the ortholog of human *MAT1A* and *MAT2A*, key genes in SAM production, and sertraline appears to induce a splicing shift that may alter SAM levels in brain cells, with effects on epigenetic regulation and neuronal function [32].

The link between *Sam-S* splicing and SAM levels is relevant to the activity of methyltransferases such as PRMT5, which depends on SAM as a cofactor [33]. *MAT2A* inhibition reduces SAM availability, impacting PRMT5 activity and gene expression. Thus, by modulating Sam-S splicing, sertraline may influence methylation processes in *Drosophila* neuronal cells, potentially affecting synaptic plasticity and the stability of neuronal connections. These effects into adulthood highlight the critical role of methylation pathways in maintaining neuronal adaptation and homeostasis.

Sam-S splice variants also play a role in cellular activity and synaptic function regulation. Variations in SAM levels, influenced by Sam-S expression and its human orthologs, impact epigenetic regulation and gene expression, directly affecting neuronal function [34,35]. For sertraline, the preference for Variant B suggests that the drug might modulate neuronal capacity for methylation and expression of genes essential for neuronal adaptation.

In summary, the sertraline-induced splicing shift in Sam-S could have significant implications for SAM regulation in the *Drosophila* brain, affecting epigenetic processes and neuronal plasticity.

The sertraline treatment induces a change in the alternative splicing of Tm1, favoring the expression of specific tropomyosin isoforms that may impact actin cytoskeleton regulation in *Drosophila melanogaster* neurons. Our results demonstrated that early larval exposure to sertraline also leads to splicing changes in tropomyosin isoforms in adulthood, highlighting the enduring effects of this drug on cytoskeletal regulation and neuronal function. Tropomyosin isoforms play a crucial role in organizing and stabilizing actin filaments, influencing key neuronal processes such as axonal and dendritic growth, morphogenesis, and synaptic plasticity [36,37].

Various tropomyosin isoforms perform specific functions in actin regulation and intracellular signaling, essential for the formation and maintenance of dendritic spines and synaptic connections [36,38]. The sertraline-induced modulation of Tm1 splicing in larval stages appears to limit the neurons’ ability to reorganize their cytoskeleton during adulthood, potentially impacting synaptic connectivity and neuronal plasticity over time. The sertraline-induced alteration in Tm1 splicing may, therefore, limit the neurons’ ability to reorganize their cytoskeleton, potentially impacting connectivity and synaptic transmission across the neural network.

Additionally, previous studies in cellular models have shown that the modulation of certain tropomyosin isoforms can alter gene expression and cellular function, particularly in neurons [39]. In this context, the shift in Tm1 splicing may contribute to the regulation of genes associated with the actin cytoskeleton, thus influencing neuronal plasticity and connectivity in *Drosophila*.

This effect on Tm1 splicing again suggests that sertraline could influence the structural and functional stability of the neuronal cytoskeleton, which may have significant implications for understanding the effects of this drug on synaptic plasticity regulation in neuron cells. Future research is warranted to explore the mechanisms through which sertraline-induced splicing changes in Tm1 influence long-term neuronal structure and behavior, and their potential relevance to therapeutic strategies in humans.

Sertraline also appears to influence mutually exclusive exon splicing in the *Tep2* gene, favoring the expression of variant B in treated samples compared to the predominance of variant A in controls. Our findings suggested long-term effects on immune-related pathways. *Tep2*, a protein containing a thioester bond, plays a critical role in the innate immunity of *Drosophila melanogaster*, facilitating the recognition and elimination of pathogens [40]. Alternative splicing of *Tep2* results in different isoforms that may have specific roles in immune response, allowing the organism to adapt to various infections and pathogens.

The shift toward variant B induced by sertraline could impact Tep2’s efficiency and specificity in pathogen recognition and response. Proteins with thioester domains, like Tep2 variants, are associated with the modulation of inflammatory response and metabolism during infections [41]. This splicing shift might suggest that sertraline indirectly influences the immune readiness of adult flies by altering immune gene isoform proportions established during larval stages, potentially optimizing responses to specific pathogens.

Further, studies have indicated that other agents, such as alcohol, also modulate alternative splicing in immune genes like *Tep2*, reinforcing the idea that splicing is an adaptive mechanism cells employ to manage environmental stress [42]. In this context, the persistent influence of sertraline on Tep2 splicing highlights its role in modulating the immune system’s adaptability to environmental and pharmacological challenges, extending beyond its serotonergic effects.

Since *Tep2* is orthologous to the human *CD109* gene and is involved in immune regulation and cell proliferation, this sertraline-induced splicing modulation may have functional parallels in more complex systems [43]. Sertraline’s impact on Tep2 splicing could, therefore, have implications for key immunological processes, such as inflammation regulation and infection response, in an evolutionary context of adaptation to adverse environmental conditions.

## 4. Materials and Methods

### 4.1. Experimental Treatment

Third-instar larvae (72 ± 4 h old) of the Canton-S WT strain of *Drosophila melanogaster* were exposed to sertraline (Pfizer, Altruline^®^, New York, NY, USA) at a concentration of 50 mg dissolved in DMSO 1%. This concentration corresponds to a standard dosage commonly used in pharmacological studies of SSRIs, including sertraline, to investigate its molecular effects and ensure adequate solubility in DMSO [44,45,46]. Negative controls included larvae treated with MilliQ water, while solvent controls were prepared with DMSO 1% (Merck, Darmstadt, Germany). The exposure period lasted 24 h under controlled environmental conditions of 25 °C, 65% relative humidity, and a 12:12 h light:dark photoperiod. Once adulthood was reached, defined as approximately 30 min after eclosion, flies were allowed an additional 4 h before collection. The adult flies were then anesthetized with cold exposure and immediately frozen in dry ice for further analysis.

### 4.2. Neural Tissue Isolation Procedure

The heads of 100 anesthetized adult males from each experimental group were dissected in a chilled PBS solution pH 7 (4 °C). The collected tissues were stored at −70 °C for subsequent RNA extraction. The entire dissection, sacrifice, and storage process was completed within 30 min to minimize potential alterations in sample quality.

### 4.3. RNA Isolation Techniques

The extraction of RNA was carried out using the RNeasy Kit (QIAGEN, Venlo, The Netherlands), as per the manufacturer’s protocol. The isolation of messenger RNA (mRNA) was further refined using the RNeasy Pure mRNA Bead Kit (QIAGEN, Venlo, The Netherlands), with quantification accomplished through the NanoDrop 200 system. All extracted samples fulfilled stringent quality standards, including a concentration of at least 5.0 ng/µL, a minimum volume of 20 µL, and an OD260/280 purity ratio equal to 2. This procedure was duplicated for each sample.

### 4.4. Bioinformatic Analysis

The transcriptomic profiling of the samples was conducted using RNAseq, set up on the Illumina NovaSeq 6000 platform, carried out by Novogene Corporation Inc. (Sacramento, CA, USA). The techniques and parameters described in this subsection for quality control, gene expression analysis, and statistical analysis are those implemented and provided by Novogene Corporation Inc.).

Following sequencing, raw data in the FASTQ format were analyzed using fastp, where reads were cleaned along with the calculation of Q20, Q30, and GC content. All subsequent analyses were based on these calculations and the clean sequences [47].

HISAT2 v2.0.5 was used to accomplish the mapping. Reference genome and gene model annotation files were downloaded directly from the genome website. An index of the reference genome was built using Hisat2 v2.0.5, and paired-end clean reads were aligned with the reference genome using Hisat2 v2.0.5. Hisat2 was selected as the mapping tool because Hisat2 can generate a database of splice junctions based on the gene model annotation file and thus a better mapping result than other non-splice mapping tools [48].

Mapped regions can be classified as exon, intron, or intergenic regions. Exon-mapped reads should be the most abundant type of read when the reference genome is well-annotated. Intron-reads may be derived from pre-mRNA contamination or intron-retention events from alternative splicing.

To quantify the baseline expression level of the transcripts, RPKM was calculated with StringTie2 [49]. Subsequently, alternative splicing events were quantified in the RNA-seq data, and the junction reads, which are essential for determining exon connectivity within a transcript, were identified and quantified using the CIGAR string [50].

Alternative splicing (AS) analysis was performed by the software rMATS (3.2.5) to analyze alternative splicing events, which mainly included three types of include exon skipping (SE), mutually exclusive exons (MXE), and retained introns (RI). Also, to quantify the proportion of exon inclusion in an alternative splicing event, the IncLevel was calculated in rMATS (3.2.5) [51].

For the analysis of alternative splicing events, a statistical model based on the edgeR package v4.0 method was applied. Events with a false discovery rate (FDR) < 0.05 were considered as differentially alternative splicing events, ensuring a reduction in false positives in the identification of significant isoform expression changes.

## 5. Conclusions

Our findings demonstrated that sertraline treatment induces significant changes in the alternative splicing of key genes associated with neuronal function in *Drosophila melanogaster*, encompassing various types of splicing events. Notably, these changes persisted into adulthood despite a single third instar larval exposure, highlighting the long-term impact of sertraline on gene expression regulation. For *Ank2* and *ATPalpha*, increased exon inclusion may contribute to enhanced axonal stability and improved neuronal functionality. In *yuri*, *sxc*, *Atg18a*, and *stmA*, changes in intron retention suggest that sertraline may modulate epigenetic processes and autophagy, both of which are critical for synaptic plasticity and cellular homeostasis. Meanwhile, *Tm1*, *gish*, *Tep2*, and *Dscam1* exhibited variations in mutually exclusive exon splicing, suggesting sertraline’s influence on mechanisms related to cytoskeletal organization, adaptive immune response, and the formation of specific neural circuits, including memory and sensory adaptation.

These findings suggested that sertraline may exert persistent neuroprotective and functional modulatory effects on the nervous system, extending beyond serotonin reuptake inhibition. The observation that larval exposure leads to lasting changes in adult gene expression emphasizes the potential translational relevance of these findings to understand sertraline’s systemic effects across life stages. However, given the invertebrate nature of our model, further studies in vertebrate models are needed to confirm these effects and assess their therapeutic potential in modulating neuronal plasticity and adaptation to stress conditions.

## Figures and Tables

**Figure 1 ijms-26-00563-f001:**
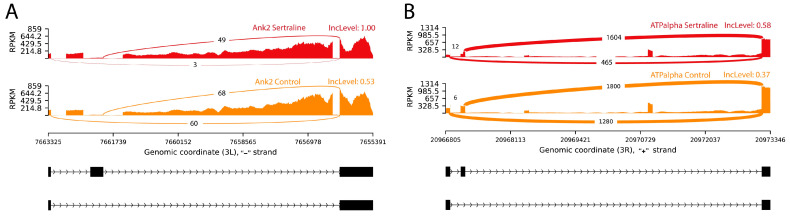
Sashimi plots illustrating statistically significant alternative splicing events by exon inclusion in the (**A**) *Ank2* and (**B**) *ATPalpha* genes. The red histograms represent RNA-seq read coverage for sertraline-treated samples, while the orange histograms correspond to control samples. The black blocks indicate annotated genomic regions, and the connecting lines represent spliced regions. For *Ank2*, sertraline-treated samples exhibited higher exon inclusion levels compared to controls, suggesting a shift towards the long *Ank2* isoform. Similarly, for *ATPalpha*, sertraline treatment promoted the inclusion of the long isoform, which is associated with enhanced neuronal functionality. Abbreviations: IncLevel—inclusion level, RPKM—reads per kilobase million.

**Figure 2 ijms-26-00563-f002:**
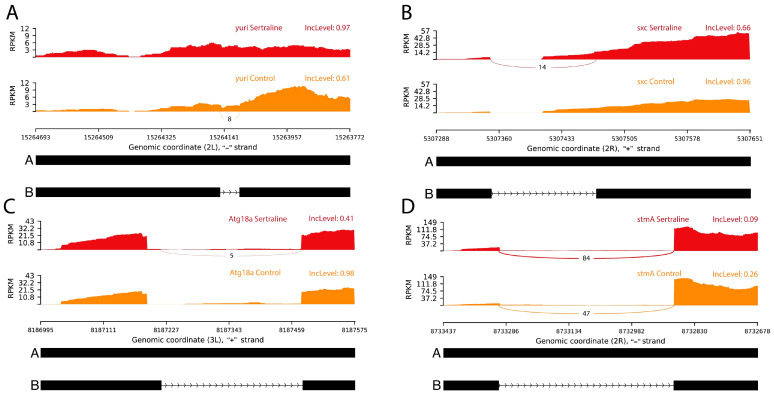
Sashimi plots displaying significant intron retention events in the genes (**A**) *yuri*, (**B**) *sxc*, (**C**) *Atg18a*, and (**D**) *stmA*. The red histograms represent RNA-seq read coverage for sertraline-treated samples, while the orange histograms correspond to control samples. The connecting lines represent spliced regions. For the *yuri* gene, intron retention was significantly higher in sertraline-treated samples compared to controls, suggesting a splicing alteration that could affect the expression of the functional isoform. For *sxc*, *Atg18a*, and *stmA*, sertraline treatment reduced intron retention, which may favor the expression of functional isoforms. Abbreviations: IncLevel—inclusion level, RPKM—reads per kilobase million.

**Figure 3 ijms-26-00563-f003:**
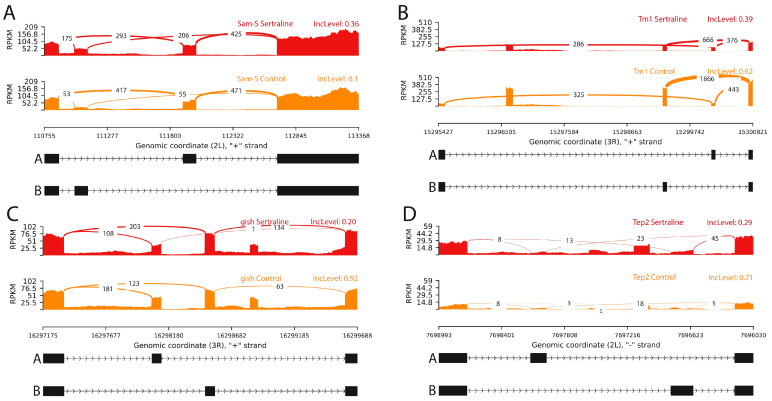
Sashimi plots displaying significant mutually exclusive exon splicing events in the genes (**A**) Sam-S, (**B**) Tm1, (**C**) gish, and (**D**) Tep2. The red histograms represent RNA-seq read coverage for sertraline-treated samples, while the orange histograms correspond to control samples. The connecting lines represent spliced regions. For Sam-S and Tm1, sertraline treatment increased the inclusion of alternative exons, while in gish, exon exclusion was observed. In Tep2, there was a higher inclusion of alternative exons following treatment. Abbreviations: IncLevel—inclusion level, RPKM—reads per kilobase million.

## Data Availability

Data are available on Figshare platform, under the link https://figshare.com/account/home#/projects/167981.

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
