# Peer review of "Impact of Larval Sertraline Exposure on Alternative Splicing in Neural Tissue of Adult Drosophila melanogaster"

_ijms, 2025, doi:10.3390/ijms26020563_

Round 1
Reviewer 1 Report
Comments and Suggestions for Authors
The study entitled: "Impact of Larval Sertraline Exposure on Alternative Splicing in 2 Neural Tissue of Adult Drosophila melanogaster" focuses on possible changes in the splicing of genes involved in structure and function of neuronal structures after treatment with the antidepressant drug sertraline. The paper is well written and the problem is addressed correctly. Some changes could make the work more complete and interesting.
Major revisions:
- Did the Canton S strain treated with sertraline show behavioral disturbances or changes in lifespan?
- Has the treatment with sertraline been tested for a longer time, for example, starting from the first larval stage?
-Were effects on other genes also observed?
- Were Drosophila melanogaster females also analyzed?
Minor revision:
In the 4.1 paragraph: "Once adulthood was reached, the adult flies were anesthetized with cold exposure and immediately frozen in dry ice for further analysis", please specify adulthood (1, 2 3 or more days after eclosion?)
How do you select the sertraline concentration to be tested?
Author Response
Major revisions:
Comment 1:
Did the Canton S strain treated with sertraline show behavioral disturbances or changes in lifespan?
Response:
Thank you for the question. In this study, we did not evaluate behavioral changes or lifespan, as our primary focus was on the effects of sertraline on alternative splicing. However, we recognize that this is an important area for future investigation to better understand the functional implications of the observed gene expression changes.
Comment 2:
Has the treatment with sertraline been tested for a longer time, for example, starting from the first larval stage?
Response:
We appreciate the reviewer’s thoughtful question. In this study, sertraline treatment was of 24 h and limited to the third instar larval stage (from 72 to 96 ± 4 hours old), to evaluate its effects on the alternative splicing of key genes in the neural tissue of adult flies. However, we agree that extending the treatment to include the first larval stage could provide valuable insights into the cumulative impact of early-life exposure to sertraline on neuronal development and splicing regulation. This approach represents an interesting avenue for future research to comprehensively explore the implications of sertraline treatment at different developmental stages.
Comment 3:
Were effects on other genes also observed?
Response:
Thank you for your observation. In our study, the genes reported were those that showed statistically significant differences in alternative splicing following sertraline treatment, based on the bioinformatics analyses performed. These included Ank2, ATPalpha, yuri, sxc, Atg18a, stmA, Tm1, gish, Tep2, and Dscam1, all of which are linked to key neuronal functions.
However, the RNA-seq dataset generated in this work contains information about many other genes that were not explored in this first phase of the analysis. We believe that a more detailed exploration of the transcriptome could identify additional effects of sertraline on genes not included in this study. This expanded analysis is among our plans for future research and could provide a more comprehensive understanding of the impact of sertraline on gene expression.
Comment 4:
Were Drosophila melanogaster females also analyzed?
Response:
In this study, we focused exclusively on male Drosophila melanogaster to avoid potential variability linked to genetic differences between sexes. In Drosophila, males have an XY chromosomal system while females are XX, which can influence gene expression and alternative splicing mechanisms through processes such as dosage compensation. This approach allowed us to isolate the effects of sertraline treatment without introducing additional variables related to sex-specific gene regulation.
However, we acknowledge the importance of including females in future studies to assess whether these differences affect the response to treatment.
Minor revision:
Comment 1:
In the 4.1 paragraph: “Once adulthood was reached, the adult flies were anesthetized with cold exposure and immediately frozen in dry ice for further analysis”, please specify adulthood (1, 2, 3 or more days after eclosion?).
Response:
Thank you for pointing this out. We have clarified that adulthood was defined as approximately 30 minutes after eclosion, allowing sufficient time for the flies’ exoskeletons to harden, their wings to expand and fill with hemolymph, and their bodies to acquire full adult coloration. Flies were collected 4 hours after this point. This clarification has been added to Section 4.1 on page 12 (lines 516–519).
Comment 2:
How do you select the sertraline concentration to be tested?
Response:
Thank you for this question. The selected concentration of 50 mg sertraline corresponds to the standard dosage commonly used in pharmacological studies of SSRIs, including sertraline, to investigate its molecular effects. This explanation, along with a reference, has been added on page 12 (lines 511–513).

Reviewer 2 Report
Comments and Suggestions for Authors
The work submitted for review, entitled "Impact of Larval Sertraline Exposure on Alternative Splicing in Neural Tissue of Adult Drosophila melanogaster," is well written, and I find the topic engaging. The authors effectively introduce the subject in the introduction, summarising it with a well-defined research hypothesis. The carefully prepared figures warrant attention, but I would appreciate more detailed information in their descriptions. Additionally, in the description of Fig. 1, you use lowercase letters (a, b), whereas the figure itself contains uppercase letters (A, B). The same applies to the subsequent figures. This inconsistency should be addressed.
Author Response
Comment 1:
The carefully prepared figures warrant attention, but I would appreciate more detailed information in their descriptions.
Response:
Thank you for this suggestion. We have revised the figure legends to include more detailed descriptions, providing additional context about the results and their significance. These updates aim to enhance the clarity and interpretability of the figures, aligning them more closely with the manuscript’s narrative. The revised figure legends can be found in the updated manuscript.
Comment 2:
In the description of Fig. 1, you use lowercase letters (a, b), whereas the figure itself contains uppercase letters (A, B). The same applies to the subsequent figures. This inconsistency should be addressed.
Response:
Thank you for pointing this out. We have revised the figure legends to ensure that all references to figure panels consistently use uppercase letters (A, B), matching the labels in the figures themselves. This correction has been applied throughout the manuscript.

Reviewer 3 Report
Comments and Suggestions for Authors
Please check attached file.

Please check comments to authors.
Author Response
Comment 1:
In the abstract, please consider mentioning a description of the splicing events.
Response:
Thank you for the suggestion. We have added a brief description of the relevance of alternative splicing in neuronal regulation and its involvement in neuronal plasticity (lines 19-22). We also specified the types of splicing events evaluated, such as exon inclusion, exon exclusion, and mutually exclusive events ( page 1, lines 23-24).
Comment 2:
In the Introduction, please consider adding a connector in the context of using model organisms to study splicing events before mentioning the Drosophila model. Review the wording and flow of the information.
Response:
Thank you for the suggestion. We have added a connector in the Introduction to establish the context of using model organisms before introducing Drosophila melanogaster. The modified paragraphs now span from page 2, line 83 to 96, improving the flow of information and connecting the ideas more coherently.
Comment 3:
In materials and methods, please review whether Laser Fluorescence Microscopy or related microscopy techniques were used to examine neuronal shapes.
Response:
Thank you for the comment. In this study, the primary focus was on analyzing changes in alternative splicing through RNA-seq, without the use of Laser Fluorescence Microscopy or related microscopy techniques to examine neuronal shapes. The data obtained were analyzed using bioinformatics methods to assess alternative splicing events following sertraline treatment. Since this work focuses on transcriptomic analysis, it would be relevant to address potential morphofunctional changes in the nervous system in future studies, given the observed impact on alternative splicing.
Comment 4:
In the presentation of results, please consider incorporating phrases that introduce the findings, mentioning the techniques and procedures used, justifying them.
Response:
Thank you for the suggestion. We have incorporated phrases introducing the techniques used, such as RNA-seq, and justifying their use in the context of the study. The modifications were made in page 3 lines 107 to 111 and 139 to 143, where we explain how these techniques were applied to identify alternative splicing events and intron retention following sertraline treatment.
Comment 5:
Please consider describing the figures justifying the claims and observations.
Response:
We appreciate the suggestion. The figure legends have been modified to include more detailed descriptions that support the claims and observations mentioned in the text. This enhances the interpretation of the presented results and their relationship to the conclusions of the study.
Comment 6:
Similarly, please consider presenting the results recorded by the techniques used to verify the findings for novice readers in this topic and area.
Response:
Thank you for the suggestion. We have added a brief explanation about IncLevel and FDR in the Results section (page 3 lines 115-120), to clarify how these metrics are used in the analysis of alternative splicing events, making them easier to understand for novice readers.
Comment 7:
In the results section, you should justify why the genes mentioned in the different splicing events were selected.
Response:
Thank you for the suggestion. The genes mentioned were not specifically selected, but instead emerged as those whose transcripts showed statistically significant splicing changes after a global analysis performed using RNA-seq. This approach allowed for the objective identification of genes with relevant changes in alternative splicing. Additionally, thanks to your previous observations, it is now clearer that the selection of the genes studied is based on the results from the global analysis of alternative splicing events. This clarification can be found in the Results section, specifically in lines 107 to 111, where we have modified the text to better reflect this approach.
Comment 8:
In the discussion of the results, protein expression in neurostructures is mentioned. In this context, and in relation to the mention of plasticity modifications; in the workflow, was any macroscopic modification observed to highlight? For example, using multiphoton microscopy techniques or any other imaging-based techniques? It is just a suggestion that could be mentioned.
Response:
Thank you for the suggestion. In this study, we focused primarily on transcriptomic analysis using RNA-seq, which allowed us to identify changes in alternative splicing of genes related to neuronal function. Although the observed changes in splicing may be related to alterations in protein expression, no direct analysis of neuronal structures or associated morphological changes was conducted. This approach was based on identifying modifications at the RNA level. As a result of your observations, we have modified the text to clarify that the relationship between protein expression changes and neuronal morphology is inferred based in published literature but was not directly addressed in this work. The modification can be found in page 7, lines 274 to 277.
